# A self-efficacy education programme on foot self-care behaviour among older patients with diabetes in a public long-term care institution, Malaysia: a Quasi-experimental Pilot Study

Siti Khuzaimah Ahmad Sharoni,[1,2] Hejar Abdul Rahman,[2] Halimatus Sakdiah Minhat,[2] Sazlina Shariff Ghazali,[3] Mohd Hanafi Azman Ong[4]

► Prepublication history and additional material are available. To view these files please visit the journal online (http://dx.doi.org/10.1136/bmjopen-2016-014393).

For numbered affiliations see end of article.

**Correspondence to**
Dr. Hejar Abdul Rahman;
hejar@upm.edu.my

## ABSTRACT

**Objective** A pilot self-efficacy education programme was conducted to assess the feasibility, acceptability and potential impact of the self-efficacy education programme on improving foot self-care behaviour among older patients with diabetes in a public long-term care institution.

**Method** A prequasi-experimental and postquasi-experimental study was conducted in a public long-term care institution in Selangor, Malaysia. Patients with diabetes aged 60 years and above who fulfilled the selection criteria were invited to participate in this programme. Four self-efficacy information sources; performance accomplishments, vicarious experience, verbal persuasion and physiological information were translated into programme interventions. The programme consisted of four visits over a 12-week period. The first visit included screening and baseline assessment and the second visit involved 30 min of group seminar presentation. The third and fourth visits entailed a 20-min one-to-one follow-up discussion and evaluation. A series of visits to the respondents was conducted throughout the programme. The primary outcome was foot self-care behaviour. Foot self-efficacy (efficacy-expectation), foot care outcome expectation, knowledge of foot care, quality of life, fasting blood glucose and foot condition were secondary outcomes. Data were analysed with descriptive and inferential statistics (McNemar's test and Wilcoxon signed-rank test) using the Statistical Package for the Social Sciences V.20.0.

**Results** Fifty-two residents were recruited but only 31 met the inclusion criteria and were included in the analysis at baseline and at 12 weeks postintervention. The acceptability rate was moderately high. At postintervention, foot self-care behaviour (p<0.001), foot self-efficacy (efficacy-expectation), (p<0.001), foot care outcome expectation (p<0.001), knowledge of foot care (p<0.001), quality of life (physical symptoms) (p=0.003), fasting blood glucose (p=0.010), foot hygiene (p=0.030) and anhydrosis (p=0.020) showed significant improvements.

**Conclusion** Findings from this pilot study would facilitate the planning of a larger study among the older population with diabetes living in long-term care institutions.

## Strengths and limitations of this study

► This is the first intervention programme addressing foot self-care behaviour based on self-efficacy constructs among older patients with diabetes living in a public long-term care institution in Malaysia.

► The self-efficacy education programme was feasible, acceptable and successful in assisting older patients with diabetes improve foot self-care behaviour and other health outcomes.

► The study highlights the need to conduct similar interventions among older patients with diabetes living in long-term care institutions in Malaysia.

► The sample is relatively small, and the intervention employed a non-randomised and non-controlled trial, hence, the trial had high potential biases.

► Some of the outcome measurements were self-reported and may lead to recall and reporting biases. Other measurements such as data from medical records, biometrics or laboratory investigations would provide additional reliable and valid results.

**Trial registration number** ACTRN12616000210471; Pre-results.

## INTRODUCTION

Diabetes is a common chronic disease affecting older people, and it is becoming a global health concern. The International Diabetes Federation reported that there are more than 134.6 million older patients with diabetes worldwide and the number is expected to increase to 252.8 million by 2035.[1] The prevalence of diabetes is expected to increase exponentially in the next 20 years for developing countries.[2] By 2030, there would be more than 82 million older patients with diabetes in developing countries.[3]

In Malaysia, the National Health Morbidity Survey reported that 15.2% (2.6 million)

have been diagnosed with diabetes.[4] The prevalence of diabetes among those aged 55 years and above was 45%.[5] Microvascular and severe late diabetic complications were reported at 75% and 25.4%, respectively.[6] The National Diabetes Registry reported that the prevalence of neuropathy, diabetic foot ulcer and amputation were at 70%, 11.1% and 11.0%, respectively. These diabetic complications have a significant impact especially for older populations.[5]

In Malaysia, the Ministry of Women, Family and Community Development abbreviated as KPWKM, manages public long-term care institutions. It was reported that 9% of the residents living in public long-term care institutions in Peninsular Malaysia have diabetes.[7] Older patients with diabetes living in the institutions receive treatment from public health clinics under the Ministry of Health. However, little is known about diabetes management among older patients with diabetes living in long-term care facilities.

Diabetes education is effective in improving foot self-care behaviour and preventing diabetic foot complications.[8] Self-care behaviour is the ability, knowledge, skills and confidence to make daily decisions.[9] Foot self-care behaviour is essential as this can improve health outcomes. However, foot self-care behaviour awareness among Malaysians with diabetes is relatively low. The National Orthopaedic Registry Malaysia reported that approximately 17.6% of patients with diabetes attended diabetic foot clinics, 22.8% kept a diabetes booklet, 23.2% applied moisturising lotion on their feet, 26.5% wore appropriate footwear and 27.3% received formal education on foot care.[10]

Self-efficacy is defined as confidence in one's ability to perform a particular behaviour, and is expected to influence the likelihood of the behavioural occurrence.[11 12] The self-efficacy theory refers to the individual's belief, feelings and their motivation. The two essential components of this theory are the expectation of the individual's ability (self-efficacy or efficacy-expectation) and determination to practise specific behaviour (outcome expectations).[13] Previous intervention studies showed significant improvements in foot self-care behaviour after the implementation self-efficacy education strategies.[14–16] To date, few published intervention studies related to self-efficacy education programs focused on foot self-care behaviour among older patients with diabetes living in a public long-term care institution in Malaysia.

## AIM
The objective of this study was to assess the feasibility, acceptability and potential impact of the self-efficacy education programme on improving foot self-care behaviour among older patients with diabetes in a public long-term care institution. The primary outcome of the study was foot self-care behaviour. Foot care self-efficacy (efficacy-expectation), foot care outcome expectation, knowledge of foot care, quality of life,

fasting blood glucose (FBG) and foot condition were the secondary outcomes.

## METHOD
### Design and sampling
This is a prequasi-experimental and postquasi-experimental study conducted in a public long-term care institution located in Selangor, Malaysia. During the period of data collection (January 2016), there was a total of 191 residents, of whom 52 had diabetes. Older patients with diabetes aged ≥60 years, Malaysian, able to communicate in Malay and independent in the activities of daily living (ADLs) were invited to participate in this study. Older patients with diabetes with cognitive impairment, psychosis, severe depression or blind, mute and deaf were excluded from this study. The principal researcher conducted a screening process prior to sample selection.

### Elements of self-efficacy in the education programme
This study incorporated self-efficacy constructs to develop the education programme (Bandura, 1977). Self-efficacy in a person can be increased through four components: (1) performance accomplishments, (2) vicarious experience, (3) verbal persuasion and (4) physiological information. Self-efficacy in diabetes management includes building new goals, starting with small steps, identifying specific needs, giving positive feedback and encouragement, and skills improvement and problem solving for respondents who are in difficult situations.[17]

In this study, self-efficacy constructs were integrated into the programme through knowledge transfer (seminar presentations and pamphlets) and self-efficacy (self-confidence) enhancement activities. To enhance the respondents' self-efficacy, they were encouraged to develop their targets, work in small, realistic steps, be more focused, and have the confidence to perform the desired behaviour. A pamphlet (symbolic modelling) was provided for self-guidance. During follow-up meetings, the respondents who were reported as unable to perform the foot self-care behaviour received specific guidance and encouragement by the principal researcher to participate in the recommended lifestyle adjustments. A sharing session with the researcher and the local nurse was conducted among respondents who had difficulty in performing the behaviour effectively. Respondents who managed to perform the behaviour effectively were appointed as mentors (live modelling) to other respondents who were not able to do so. The respondents received weekly visits by the local nurse for physical and emotional support.

### Intervention module
Knowledge transfer was conducted during the health education programme. The respondents received information from PowerPoint presentations (PPT) (oral and visual information) and pamphlets (written and visual information). The programme was conducted in the meeting room, consisted of 10–11 respondents per group

per session and was facilitated by the researcher. The education activities included a 20-min PPT seminar aided by a pamphlet. The topic of this intervention programme was 'Foot self-care for older diabetics'. The content and design of the PPT and pamphlet were adapted from standard diabetes organisations.[1 18 19] The content of the PPT and the pamphlet highlighted the risk factors of diabetes foot complications, foot self-examination, daily foot hygiene and cleanliness, foot protection and prevention of foot-related complications.

A foot care package consisting of a pamphlet on foot self-care, a nail clipper, a water-based lotion and a small towel was given to each respondent after the seminar. A reminder checklist was developed for the local nurse in charge of the clinic at the institution. The nurse also received instructions from the researcher to remind, give support and guidance to the respondents on performing the foot self-care behaviour daily. The respondents were advised to ask for guidance from the local nurse or their colleagues. The nurse was required to put her signature on the respondent's name column of the reminder checklist after visiting the respondents.

### Variables and measurements

The feasibility of this study was measured by the respondents' recruitment, attendance, attrition and compliance rates.[20] The acceptability profile was evaluated after completing the 12-week programme with a modified version of the Abbreviated Acceptability Rating Profile.[21] Eight items were used to assess respondents' acceptability of the self-efficacy education programme. A 5-point Likert Scale (strongly disagree (1), disagree (2), neither disagree or agree (3), agree (4) and strongly agree (5)) was used. The score ranged from 8 to 40 and a higher score indicated better acceptability towards the programme delivered.

A questionnaire was used to evaluate the impact of the intervention on the outcome measures. The demographic data consisted of age, gender, ethnicity, education level, marital status, having children and duration of stay in the institution. FBG, diabetes duration, treatment of diabetes, comorbidity, previous diabetes education received, current smoking status and current history of hospitalisation related to diabetes were assessed for clinical characteristics.

The primary outcome of this study was foot self-care behaviour measured using the modified version of Diabetes Foot Self-Care Behaviour Scale (DFSBS).[22] The original DFSBS had good validity and reliability (Cronbach's α=0.73), and the test-retest reliability was 0.92.[22] Foot self-care behaviour consisted of two parts. In part 1, seven items were asked about how many days the respondents performed foot self-care behaviour in the past 7 days (1 week). Part 2 (nine items) was about the frequency in which respondents performed a certain foot self-care behaviour. The responses were rated as a 5-point Likert Scale (never/0 day per week (1), rarely/1–2 days per week (2), sometimes/3–4 days per week (3), often/5–6

days per week (4) and always/7 days per week (5)).[22] The score ranged from 16 to 80; a higher score indicated good foot self-care behaviour.

For foot care self-efficacy (efficacy-expectation), the modified version of the Foot Care Confidence Scale (FCCS) was used.[23] The reliability test of the original FCCS was high (Cronbach's α=0.92).[23] The foot care self-efficacy (efficacy-expectation), consisted of 10 items with a 5-point Likert Scale (strongly not confident (1), not confident (2), moderately confident (3), confident (4) and strongly confident (5)). The score ranged from 10 to 50; a higher score indicated higher confidence to perform the foot self-care behaviour.

Foot care outcome expectation was developed based on previous literature.[12 23–25] The scale consisted of six items with a 5-point Likert Scale (strongly disagree (1), disagree (2), neither disagree or agree (3), agree (4) and strongly agree (5)). The score ranged from 6 to 30 and a higher score indicated that the respondents had higher confidence in performing good foot self-care behaviour.

Knowledge of foot care was developed based on previous literature.[26–28] The instrument assessed the respondents' knowledge about risk factors, common diabetic foot complications and foot care. The scale consisted of 11 items with three possible answers (true, false and don't know). One mark was given for each correct answer. The score ranged from 0 to 11. A higher score indicated a higher level of knowledge.

The modified version of the Neuropathy and Foot Ulcer Specific Quality of Life was used to assess the quality of life of the respondents.[29] The original instrument showed a good reliability (Cronbach's α=0.86–0.95). The instrument was divided into physical symptoms (13 items) and psychosocial functioning (12 items). This is an ordinal scale divided into two sections. First, the respondents need to respond to the foot problems affecting their well-being (always (3), sometimes (2), never (1)) and psychosocial functioning (every time (3), seldom (2), none (1) or agree (3), neither agree or disagree (2) and disagree (3)). In the second section, the respondents were asked about whether the foot problems bother them (none (1), some (2), or very (3)). The total score was calculated by multiplying the score in section 1 and section 2. The score for physical symptoms ranged between 13 and 117 and for psychosocial functioning the score range was from 12 to 108. A lower score indicated a better quality of life.

Foot condition was developed from previous literature.[30 31] The instrument assessed overall hygiene, nail conditions, common skin conditions, other conditions and infections and complication. If respondents had a foot condition, it was rated as '1 point' for each component (can be either left or right).

### Validity and reliability

The intervention module and the instruments were assessed for face and content validity. This was to ensure that the materials were appropriate for current local practice, culturally equivalent, and appropriate for the

population and study location. Content validity ratio (CVR) of the instruments was assessed by six experts; one endocrinologist, two public health specialists, one family medicine specialist, one diabetic nurse educator and one older diabetic.

The formula CVR = $(2\,ng\,/\,N)-1$, was used for validity conformity[32] where ng is the number of panel experts, who thinks that the item is good, and N is the total number of panel experts. In this approach, six panel experts were asked to indicate if a measurement item in a set of items is 'essential' or 'useful but not essential' or 'not necessary' to the operationalisation of a theoretical construct.[33]'Essential' items or assessment tasks are ones that best represent the goal and are desired. The value lies between −1.00 to +1.00, where CVR=0.00 means that 50% of the panel experts in a panel size of N believe that the portfolio task is essential and therefore valid (Johnston & Wilkinson, 2009).[33] In this study, the items were excluded when CVR <0.00.

Forward and backward translations from English to Malay and back to English was performed by two bilingual translators certified by the Institute of Language and Literature Malaysia. The second draft of the questionnaire was weighted carefully before data collection commenced.

The results of internal consistency or reliability tests were acceptable for foot self-care behaviour (Cronbach's α=0.68), foot care self-efficacy (efficacy-expectation) (Cronbach's α=0.91), foot care outcome expectation (Cronbach's α=0.88), knowledge of foot care (Cronbach's α=0.86) and quality of life (physical symptoms, Cronbach's α=0.68) (psychosocial functioning, Cronbach's α=0.68).

Data collection procedures were conducted by the principal researcher (registered nurse), a local nurse (community nurse in charge of the clinic) and a research enumerator (registered nurse). The research enumerator received 2 hours of training with a manual file. The local nurse received a 30-min briefing by the principal researcher regarding visiting procedures and checklist usage.

## Data collection procedures

The data collection process consisted of four visits. During the first visit, the activities included a screening session of all older patients with diabetes for inclusion and exclusion criteria, consent taking procedures and baseline assessments. The respondents received information about the study's objectives, procedures, benefits and potential harms from the main researcher. Data collection was conducted by a trained research enumerator from baseline to week 12.

The second visit was a 30-min seminar presentation delivered by the principal researcher in a meeting room at the institution. The respondents were divided into three groups: group A (10 respondents), group B (10 respondents) and group C (11 respondents). The presentation was delivered three times (one group/session). At the end of the seminar, each respondent received a foot care package. The local nurse conducted a weekly (between week 0 and week 4) visit to the respondents.

The third visit (week 4) was for follow-up and problem solving. A 20-in session of one-to-one discussion sought to identify any obstacles and provided personal feedback and positive support. A special meeting was held between the principal researcher, the local nurse and respondents who had little experience in performing the foot self-care behaviour effectively (as reported by the local nurse). A respondent who could perform the effective behaviour was appointed as a mentor for skill improvement. Biweekly, (between week 5 and week 12) the local nurse visited the respondents between the third and fourth visits.

At week 12, the respondents were evaluated for outcome measures by the research enumerator. The respondents were evaluated for acceptability, compliance, effectiveness and maintenance towards this programme. Continuous support and strong encouragement were given so they would perform the behaviour regularly. The local nurse was asked to share their experience regarding any limitations and to offer suggestions to improve the programme.

## Ethical procedures

This study has been approved by the Universiti Putra Malaysia and the Social Welfare Department, Malaysia. The respondents were briefed on the study using the subject information sheet and consent was obtained prior to data collection. The data were treated as private and confidential.

## Statistical analysis

Data were analysed with descriptive and inferential statistics using the Statistical Package for the Social Sciences (SPSS) V.20.0. Demographic data, clinical characteristics and acceptability profile were presented in mean and SD or frequency (n) and percentage (%). Since the sample was small, non-parametrical McNemar's test and Wilcoxon signed-rank test were used to assess the outcomes.

## RESULTS
### Respondents' characteristics and feasibility of the study

There were 52 older residents with diabetes. One of them refused to participate due to a non-specific reason. Therefore, 51 were screened for eligibility and 21 were excluded. The reasons for exclusion included; bedridden (5), cognitively impaired (6), known dementia (3), language barrier (2), depressed (1), known psychosis (2), deaf/mute (1) and blind (1). Hence, 31 respondents (60.8%) were eligible and agreed to participate (see figure 1). For the intervention attendance, all respondents (n=31) were involved in the seminar presentation and week-4 follow-up (100%). The attrition rate in this study was 0.0% at week-4 to week-12. The final number at week-12 included 31 respondents.

On average, the age of respondents was 69 years (SD=4.23), most were female (54.8%), Malay (58.1%),

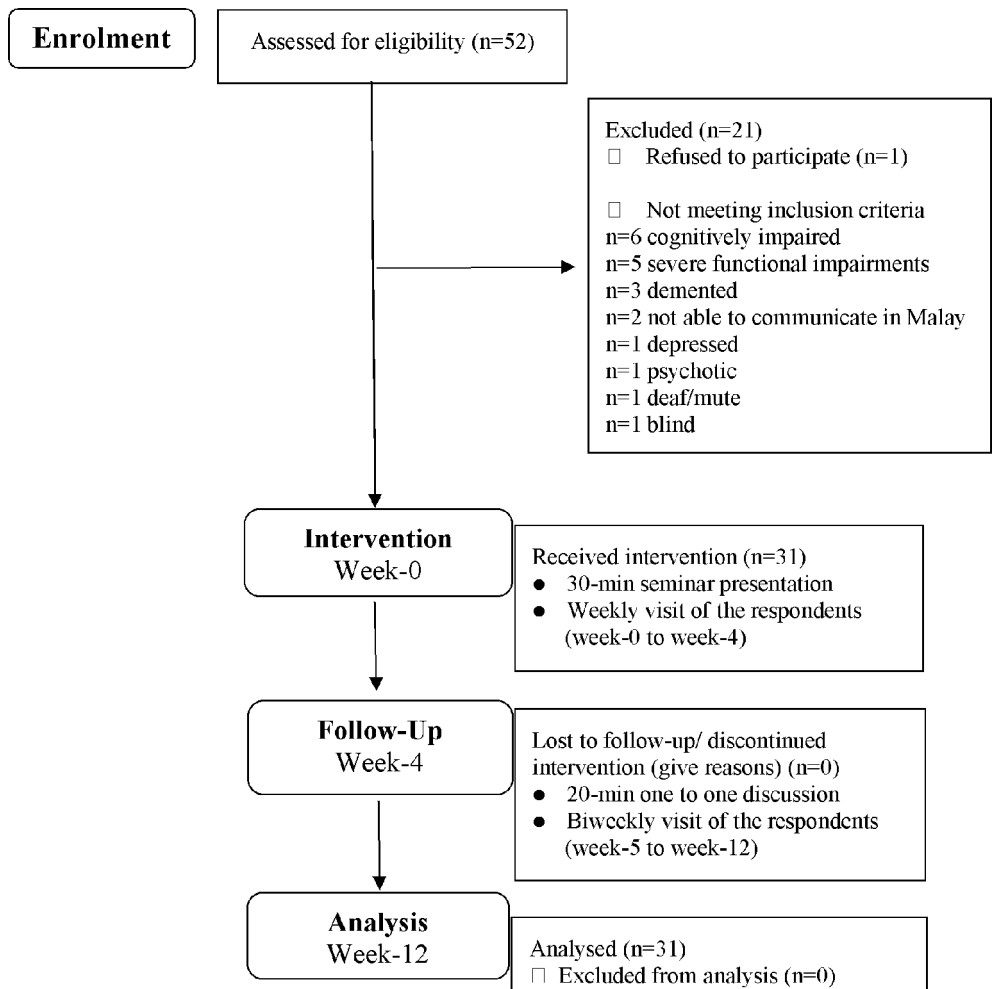

**Figure 1** Flow diagram of enrolment , intervention, follow-up and analysis of the study. The figure was modified from the CONSORT 2010 flow diagram.

attended primary school (51.6%), married (61.3%) and did not have children (61.3%). The average duration of the respondents living in the institution was 5 years (SD=3.84) (see table 1).

On average, the respondents' FBG was 8.66 mmol/L (SD=3.15) and had been diagnosed with diabetes for 12 years (SD=12.95). Most of them were on oral medication(s) (74.2%), had comorbid disease(s) (93.5%) and had never received any diabetes education (71.0%); all of them had no history of hospitalisation related to diabetes in the 3 months before data collection (100.0%) and 54.8% of them were non-smokers (see table 2).

The recruitment rate and retention rate was 100% as all respondents were enrolled and completed the 12-week programme. Figure 1 shows the flow diagram of enrolment, intervention, follow-up and analysis of the study. The figure was modified from the CONSORT 2010 flow diagram.[34]

### Acceptability of the study

On average, the acceptability score was moderately high (mean=33.84±4.08). A majority of the respondents reported that the programme was acceptable (mean=4.32±0.48), effective (mean=4.06±0.81) and can be applied to other older patients with diabetes (mean=4.29±0.46). The respondents liked this programme, considered the programme a good way to prevent diabetic foot problems (mean=4.32±0.48), found it helpful and had no adverse effects. However, the respondents were unsure if they will continue to perform the behaviour after this programme (mean=3.87±0.81) (see table 3).

### Foot self-care behaviour, foot care self-efficacy (efficacy-expectation), foot care outcome expectation, knowledge of foot care, quality of life and FBG

The normality of the continuous data was assessed with the Shapiro-Wilk test (table 4). The analysis indicated that a majority of the variables were normally distributed as the Shapiro-Wilk test was significant (p<0.05). Therefore, a non-parametrical test (ie, Wilcoxon signed-rank test) was the preferred test for assessing the significant difference between pre (ie, baseline) and post (ie, evaluation) variable score test.

Table 5 shows that foot self-care behaviour levels significantly increased from the baseline assessment

**Table 1** Distribution of respondents according to demographic characteristics (n=31)

| Variables | | n | % |
|---|---|---|---|
| Age | Mean±SD=68.52 (4.23) | | |
| Gender | Male | 14 | 45.2 |
| | Female | 17 | 54.8 |
| Ethnicity | Malay | 18 | 58.1 |
| | Chinese | 3 | 9.7 |
| | Indian | 9 | 29.0 |
| | Others | 1 | 3.2 |
| Education level | Never | 4 | 12.9 |
| | Primary | 16 | 51.6 |
| | Secondary | 9 | 29 |
| | Tertiary | 2 | 6.5 |
| Marital status | Single | 12 | 38.7 |
| | Married | 19 | 61.3 |
| Having children | No | 19 | 61.3 |
| | Yes | 12 | 38.7 |
| Duration of stay | Mean±SD=4.84 (3.84) | | |

**Table 2** Distribution of respondents according to clinical data (n=31)

| Variables | | n | % |
|---|---|---|---|
| Fasting blood glucose (FBG) | Mean±SD=8.66 (3.15) | | |
| Diabetes duration | Mean±SD=11.81 (12.95) | | |
| Treatment | Oral | 23 | 74.2 |
| | Insulin | 2 | 6.5 |
| | Oral and insulin | 6 | 19.4 |
| Comorbidity | No | 2 | 6.5 |
| | Yes | 29 | 93.5 |
| Had received previous diabetes education | No | 22 | 71.0 |
| | Foot care | 1 | 3.2 |
| | Others (eg, diet, exercise, medication, blood glucose monitoring, smoking cessation) | 8 | 25.8 |
| Recent hospitalisation (DM) | No | 31 | 100.0 |
| | Yes | 0 | 0.0 |
| Current smoking | No | 17 | 54.8 |

DM, diabetes mellitus.

**Table 3** The acceptability profile to the programme delivered (n=31)

| Variables | Mean±SD |
|---|---|
| This is an acceptable programme for you | 4.32±0.48 |
| The programme should be effective in changing the foot self-care behaviour | 4.06±0.81 |
| This programme can be used for other older patients with diabetes who did not perform foot self-care behaviour properly | 4.29±0.46 |
| You will continue to perform the foot self-care behaviour after this programme | 3.87±0.81 |
| This programme would not have bad side effects for you | 4.32±0.48 |
| You liked this programme | 4.32±0.48 |
| The programme was a good way to prevent diabetic foot problems | 4.32±0.48 |
| Overall, the programme would help you | 4.32±0.48 |
| Total score | 33.84±4.08 |

(median=45.00) to the evaluation test (median=69.00), Z=−4.86, p<0.001. Besides that, the Wilcoxon signed-rank test analysis also indicated that foot care self-efficacy (median=30.00), foot care outcome expectation (median=19.00) and knowledge of foot care scores (median=8.00) statistically increased from the baseline test to the evaluation test (foot care self-efficacy: median=44.00, Z=−4.76, p<0.001; foot care outcome expectation: median=25.00, Z=−4.79, p<0.001; knowledge of foot care score: median=11.00, Z=−4.47, p<0.001).

The analysis reported in table 5 also indicated that compared with the baseline test, the score of quality of life for physical symptoms (median=23.00) and FBG (median=7.90) statistically decreased in the evaluation test (quality of life for physical symptoms: median=14.00, Z=−2.99, p=0.003; FBG: median=6.10, Z=−2.57, p=0.010). Meanwhile, the score of quality of life for psychosocial functioning showed no significant difference between

**Table 4** Normality assessment (n=31)

| | Shapiro-Wilk test | |
|---|---|---|
| Variables | Baseline | Week 12 |
| Foot self-care behaviour | 0.961 | 0.888* |
| Foot care self-efficacy (efficacy expectation) | 0.986 | 0.927* |
| Foot care outcome expectation | 0.955 | 0.872* |
| Knowledge on foot care | 0.826* | 0.759* |
| Quality of life | | |
| Physical symptoms | 0.834* | 0.746* |
| Psychosocial functioning | 0.737* | 0.936 |
| Fasting blood glucose (FBG) | 0.880* | 0.922* |

*p<0.05

**Table 5** Changes in the foot self-care behaviour, foot care self-efficacy (efficacy expectation), foot care outcome expectation, knowledge on foot care and quality of life before and fasting blood glucose (FBG) after the programme (n=31)

| Variables | Mean±SD (median) | | Wilcoxon signed-rank test (Z) |
| --- | --- | --- | --- |
| | Baseline | Week 12 | |
| Foot self-care behaviour | 47.00±9.21 (45.00) | 68.00±6.23 (69.00) | Z=−4.86, p=0.001* |
| Foot care self-efficacy (efficacy expectation) | 29.90±8.68 (30.00) | 43.68±4.94 (44.00) | Z=−4.76, p=0.001* |
| Foot care outcome expectation | 19.58±4.26 (19.00) | 25.97±3.43 (25.00) | Z=−4.79, p=0.001* |
| Knowledge on foot care | 6.68±2.90 (8.00) | 9.97±1.35 (11.00) | Z=−4.47, p=0.001* |
| Quality of life | | | |
| Physical symptoms | 27.48±16.65 (23.00) | 19.90±12.32 (14.00) | Z=−2.99, p=0.003* |
| Psychosocial functioning | 30.84±25.75 (15.00) | 27.58±6.72 (26.00) | Z=−0.31, p=0.754 |
| FBG | 8.66±3.15 (7.90) | 6.98±2.18 (6.10) | Z=−2.57, p=0.010* |

*p<0.05

the baseline test (median=15.00) and the evaluation test (median=26.00).

### Foot condition

Table 6 shows that at baseline, most respondents had good foot hygiene (77.4%). The most common foot condition was anhydrosis (61.3%) followed by skin fissures (29.0%), corns and calluses (25.8%), nail infection (19.3%), skin injury (16.1%), involuted nail plates (12.9%) and dermatitis (6.5%). Only 3.2% of the respondents had ingrown toenails, subungual lesion, interdigital maceration, ulcers and an amputated foot. Foot conditions improved

significantly for overall foot hygiene (p=0.03) and anhydrosis (p=0.02) after the education programme.

### DISCUSSION

This study was conducted to determine the feasibility, acceptability and potential health-related impact of the foot self-care behaviour programme. The rate of enrolment, attendance to the programme and compliance are common indicators for assessing the feasibility of such a study.[31] The rate of enrolment was moderate (60.8%). The enrolment rate of the current study (60.8%) was

**Table 6** Changes in the foot score before and after the programme (n=31)

| Variables | | n (%) | | McNemar's test |
| --- | --- | --- | --- | --- |
| | | Baseline | Week 12 | |
| Overall foot hygiene (clean, short nail) | Yes | 24 (77.4) | 28 (90.3) | 0.03* |
| Nail conditions | | | | |
| Nail infection | Yes | 6 (19.3) | 4 (12.9) | 0.63 |
| Ingrown toenail/s | Yes | 1 (3.2) | 0 (0.0) | 1.00 |
| Subungual lesion/s | Yes | 1 (3.2) | 0 (0.0) | 1.00 |
| Involuted of nail plate/s | Yes | 4 (12.9) | 4 (12.9) | 1.00 |
| Common skin conditions | | | | |
| Corns and/or callous | Yes | 8 (25.8) | 4 (12.9) | 0.22 |
| Skin fissures | Yes | 9 (29.0) | 5 (16.2) | 0.34 |
| Anhydrosis | Yes | 19 (61.3) | 10 (32.3) | 0.02* |
| Skin injury/ cuts/ abrasions/ blisters | Yes | 5 (16.1) | 3 (9.7) | 0.73 |
| Other conditions and infections | | | | |
| Dermatitis/eczema/psoriasis | Yes | 2 (6.5) | 1 (3.2) | 0.50 |
| Interdigital maceration | Yes | 1 (3.2) | 1 (3.2) | 1.00 |
| Complications | | | | |
| Ulcer/s and amputations of toes | Yes | 1 (3.2) | 1 (3.2) | 1.00 |

*p<0.05

lower when compared with another study (≥80%) on an education programme that enhances self-care practices among Malaysian adults with diabetes.[35] This disparity could be due to the difference in the inclusion criteria, where our study focused on an older population living in a long-term care institution while the other study involved adults living in the community. Our results showed that the rate of intervention attendance was high (100%), with no attrition among respondents, providing evidence that the programme was feasible. This finding was similar to a theory-based pilot study on diabetes education conducted in peninsular Malaysia among patients without diabetes, aged ≥18 years and having at least secondary level of education.[36] They reported a response rate of 96.7% and all the respondents completed both pretest and post-test assessments.

The acceptability score was highly acceptable, and the finding was similar to a pilot study conducted on foot self-care educational intervention among patients with diabetes in Canada.[31] The majority of the respondents in our study reported that the programme was effective, beneficial, and enjoyable and perceived this programme as a good way to prevent diabetic foot problems.

The study found improvements in the foot self-care behaviour, foot care self-efficacy (efficacy-expectation), foot care outcome expectation, knowledge of foot care and quality of life (physical symptoms) following the programme. Foot self-care behaviour improved after 12 weeks following the education programme. The findings were in line with previous interventional studies conducted on foot self-care among the older population with diabetes.[16 37–39]

The results showed that foot self-efficacy (efficacy-expectation) scores improved after implementing the education programme. Previous studies demonstrated similar findings such as improvements in the foot self-efficacy scores before and after the implementation of the foot self-care intervention programme.[14 15] The respondents in our study reported being more confident in undertaking foot self-care behaviour after the education programme. Self-efficacy enhancing activities were integrated throughout the data collection process; starting from the seminar presentation, follow-up session and weekly reinforcement.

The foot care outcome expectation scores also improved from baseline to 12 weeks after the programme. Likewise, the score for the outcome expectation improved after the implementation of a self-efficacy intervention for patients with diabetes, as found in another study.[40] According to Bandura, outcome expectation is about a person's belief in achieving positive outcomes when he/she performs a given behaviour.[11]

The self-efficacy education programme showed effectiveness in increasing knowledge of foot care at week 12 of the intervention. Similar to previous studies, it demonstrated that knowledge level improved after the foot self-care education programme. Diabetic foot self-care knowledge scores increased at 3 months,[15 37 41] 6 months[37]

and 8 months[42] after intervention in previous studies. At baseline, the respondents in our study had a moderate knowledge of risk factors for diabetic foot complications and foot self-care. Therefore, it can be suggested that older patients with diabetes in the institution need further information regarding the risks of diabetic foot complications and preventive measures.

There was a significant improvement in the respondents' quality of life for physical symptoms but not for psychosocial functioning. This finding was similar to Aiken's study.[43] In contrast, Williams's study stated that psychological well-being showed an improvement from baseline to 3 months postintervention but not for physical health.[41] In another study, the intervention did not lead to a significant effect on the quality of life.[44] This is difficult to explain as the dimension of the quality of life is broad and concerns many aspects such as health status, socioeconomic, culture, environment and the spiritual.[45] The differences of quality of life in this study with other studies could be due to different study locations and populations. As this study was conducted in a long-term public institution, other factors might influence the quality of life. For example, functional limitation in performing ADLs independently, disturbances in social relationship and disruption in emotional states among the residents and staff in the institution could influence psychosocial functioning.

A significant reduction in the FBG level in this study is encouraging, and this finding was similar to another Malaysian study that demonstrated the effects of self-management for patients with diabetes by using the self-efficacy concept in their education programme.[35]

The results showed improvements in overall foot hygiene and anhydrosis after the education programme, similar to Fan's study.[31] Other variables (ie, nail conditions/infections, corn or callous, skin fissure, skin injury and dermatitis) improved, but this change was not statistically significant. Likewise, there were no significant differences found in ulcer incidence and amputation at 6 months and 12 months postintervention.[39] Fujiwara's study reported that callous grade improved at 2 years postintervention.[46] The differences in the outcome measures that influenced the effects of intervention could be due to the difference in the methodological approaches, sampling selection, instruments used, study duration or clinical practice.

The intervention module used in this programme was designed from diabetes standard practice guidelines and was carefully weighted for older patients with diabetes. The programme activities, incorporated with the four self-efficacy components, were practical and acceptable for daily local practice. The pamphlet and foot care package provides the opportunity for them to carry out self-revision and individual practice. Seminar presentations and the series of follow-ups with a reminder checklist and reinforcements demonstrated a significant effect in assisting the older patients with diabetes perform foot self-care behaviour regularly. It was supported by recent clinical guidance and guidelines that diabetic foot

problems can be prevented with foot care education, foot protection and therapeutic footwear as well as professional foot examination.[47 48]

The strength of this study is that the older patients with diabetes lived together in an institution and the regular visits by the local healthcare provider allowed discussions on the process of foot self-care behaviour modification. The respondents were able to share their experience of the programme and support each other for sustainability. It is hoped that in the future, the findings of this pilot study could contribute to the implementation of a foot self-care behaviour education programme based on the self-efficacy theory. Similar education programs involving more samples of older patients with diabetes in public long-term care institutions are needed in the Malaysian context. Continuous support from the university and the Ministry of Women, Family and Community Development with the necessary resources to assist older patients with diabetes can improve the health status.

## Limitations of this study

There were several limitations in this study. It employed a non-randomised controlled trial (RCTs), and since there was no control group, there was a high risk of potential biases. The sample was relatively small, and a control group was not available to determine the effectiveness of study between groups. The lack of glycosylated haemoglobin measurements may lead to insufficient information to determine the amount of respondents' blood glucose level for the previous 3 months. Self-reports on the outcome measurements may lead to recall and reporting biases. Other types of measurements such as data from medical records, biometrics or laboratory investigations would provide additional reliable and valid results. As this study was conducted among older patients with diabetes living in a long-term care institution, the findings could not be generalised to other populations.

## CONCLUSION

These findings showed that the programme is feasible, acceptable and effective in improving foot self-care behaviour of older patients with diabetes. Based on these findings, an education programme based on the self-efficacy theory would help and facilitate the planning of a study in a larger older population with diabetes living in long-term care institutions using RCTs.

## Author affiliations
[1]Department of Nursing, Faculty of Health Sciences, Universiti Teknologi MARA, Puncak Alam, Selangor, Malaysia
[2]Department of Community Health, Faculty of Medicine and Health Sciences, Universiti Putra Malaysia, Serdang, Selangor, Malaysia
[3]Department of Family Medicine, Faculty of Medicine and Health Sciences, Universiti Putra Malaysia, Serdang, Selangor, Malaysia
[4]Department of Statistics, Faculty of Computer Science and Mathematics, Universiti Teknologi MARA, Segamat, Johor, Malaysia

**Acknowledgements** The authors thank the Universiti Teknologi MARA and the Ministry of Higher Education (MOHE) for the PhD scholarship, the Community Health Department, the Research Ethic Committee and the Putra Grant (no. 9463500). The authors also thank Dr Muhammad Mikhail Joseph Anthony Abdullah (Endocrinologist, Medical Department), Ms Nuraisyah Hani Zulkifley (Registered Nurse/ Masters student of Community Health Department) and Ms Nadia Alhana Arifin (Diabetic Nurse Educator, Family Medicine Department) Universiti Putra Malaysia, the Social Welfare Department, Malaysia and the nurses (Ms Nur Aidafitri Azahar, Ms Norfarahida Pami, Ms Norsyahida Ramli, Ms Sharifah Shahida Syed Azahar and Ms Azma Zain) involved in this study, the institution, and lastly the older patients with diabetes who participated in the study.

**Contributors** Study design: SKAS, HAR, HSM, SSG. Data collection: SKAS, HAR, SSG. Data analysis: SKAS, MHAO, HAR, SSG. Manuscript preparation: SKAS, HAR, HSM, SSG, MHAO.

**Competing interests** None declared.

**Ethics approval** Universiti Putra Malaysia and Social Welfare Department Malaysia.

**Provenance and peer review** Not commissioned; externally peer reviewed.

**Data sharing statement** Our study protocol is freely available via (https://www.anzctr.org.au/Trial/Registration/TrialReview.aspx?id=369488).

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
