## [Reviewer comments · BMJ Open]

ARTICLE DETAILS

TITLE (PROVISIONAL)	A Self-Efficacy Education Program on Foot Self-Care Behaviour among Older Diabetics in a Public Long-Term Care Institution, Malaysia: A Quasi-Experimental Pilot Study
AUTHORS	Sharoni, Siti Khuzaimah Ahmad; Abdul Rahman, Hejar; Minhat, Halimatus Sakdiah; Shariff Ghazali, Sazlina; Azman Ong, Mohd Hanafi

VERSION 1 - REVIEW

REVIEWER	Ligia de Loiola Cisneros Universidade Federal de Minas Gerais Brazil
REVIEW RETURNED	15-Oct-2016

GENERAL COMMENTS	Abstract: you must add more information about the intervention References: I would like to suggest these recent guidance and guideline (based on systematic reviews) that could be helpful for the discussion section: http://onlinelibrary.wiley.com/doi/10.1002/dmrr.2694/abstract;jsessionid=1CEB348B4311DD9F0AA521A335E09A95.f03t01 http://www.jvascsurg.org/article/S0741-5214(15)02025-X/abstract Discussion: Your results were compared to other from the literature, more than be discussed. I am sure you can make it better demonstrating your experience. Conclusion: try to restrict the conclusion to your initial question. I think the first sentence is enough. Supplementary files: At figure 1 if you present the intervention (what was done at each week) it would be easier to understand the program proposed (and assessed)
---

REVIEWER	Perry Mayer, MB, BCh, CCFP The Mayer Institute Canada
REVIEW RETURNED	29-Oct-2016

GENERAL COMMENTS	There are a few typos that need to be cleaned up. Otherwise an interesting and relevant paper that merits publication and further expanded research.
--

REVIEWER	Faraja Chiwanga Muhimbili National Hospital, Tanzania
REVIEW RETURNED	12-Dec-2016

GENERAL COMMENTS	Overall, a very good manuscript with detailed description of methodology and results I have only minor suggestions Consider separating ulcers and amputation (table 6), there is only one event at baseline and at week-12, it important to know whether this was an ulcer or amputation. Few typographical errors that need correction, I have cited a few (pg 15 line 28; pg 16 line 11)
--

VERSION 1 – AUTHOR RESPONSE

Issues raised for corrections (Reviewer 1)

5.Abstract: you must add more information about the intervention

Response: thank you for useful comments. We have made the correction accordingly under “abstract”.

6.References: I would like to suggest these recent guidance and guideline (based on systematic reviews) that could be helpful for the discussion section:

<http://onlinelibrary.wiley.com/doi/10.1002/dmrr.2694/abstract;jsessionid=1CEB348B4311DD9F0AA521A335E09A95.f03t01>

[http://www.jvascsurg.org/article/S0741-5214\(15\)02025-X/abstract](http://www.jvascsurg.org/article/S0741-5214(15)02025-X/abstract) Thank you for useful comments.

Response: we have added the guidance and guideline (based on systematic reviews) from Hingorani, et al. (2016) and Bus et al. (2015) in our study, under “discussion”, tenth paragraph, last sentence.

7.Discussion: Your results were compared to other from the literature, more than be discussed. I am sure you can make it better demonstrating your experience.

Response: thank you for useful comments. We have made the correction accordingly under “discussion”, tenth and eleventh paragraph.

8.Conclusion: try to restrict the conclusion to your initial question. I think the first sentence is enough.

Response: thank you for useful comments. We have made the correction accordingly, focused to our initial question

9.Supplementary files: At figure 1 if you present the intervention (what was done at each week) it would be easier to understand the program proposed (and assessed)

Response: thank you for useful comments. We have made the correction accordingly at supplementary file “Figure 1”.

Issues raised for corrections (Reviewer 2)

10.There are a few typos that need to be cleaned up. Otherwise an interesting and relevant paper that merits publication and further expanded research.

Response: we have asked our English editor to check the language structure. We believe that the structure and the language are now acceptable for the review process.

Issues raised for corrections (Reviewer 3)

11.Overall, a very good manuscript with detailed description of methodology and results

I have only minor suggestions

Consider separating ulcers and amputation (table 6), there is only one event at baseline and at week-12, it important to know whether this was an ulcer or amputation.

Response: thank you for useful comments. For your information, there was only one patient who had both ulcer and amputations of toes at baseline and week-12

12.Few typographical errors that need correction, I have cited a few (pg 15 line 28; pg 16 line 11)

Response: We have asked our English editor to check the language structure. We believe that the structure and the language are now acceptable for the review process.

VERSION 2 – REVIEW

REVIEWER	Faraja Chiwanga Muhimbili National Hospital Tanzania
REVIEW RETURNED	03-Feb-2017

GENERAL COMMENTS	1. Improved manuscript however still with few typos e.g title of table 62. Table 6: You are looking at the change of foot score and are looking at specific parameters; you will not be expecting to see number of amputation decrease, however, you may see a decrease in foot ulcers. So in your follow-on study with large sample size this should be separated.
--